# Self-Triggered Thermomechanical Metamaterials with Asymmetric Structures for Programmable Response under Thermal Excitations

**DOI:** 10.3390/ma14092177

**Published:** 2021-04-23

**Authors:** Pengcheng Jiao, Luqin Hong, Jiajun Wang, Jie Yang, Ronghua Zhu, Nizar Lajnef, Zhiyuan Zhu

**Affiliations:** 1Institute of Port, Coastal and Offshore Engineering, Ocean College, Zhejiang University, Zhoushan 316021, China; lqhong@zju.edu.cn (L.H.); 3170100248@zju.edu.cn (J.W.); j.yang@zju.edu.cn (J.Y.); zhu.richard@zju.edu.cn (R.Z.); 2Engineering Research Center of Oceanic Sensing Technology and Equipment, Zhejiang University, Ministry of Education, Hangzhou 310027, China; 3Hainan Institute of Zhejiang University, Sanya 572025, China; 4Department of Civil and Environmental Engineering, Michigan State University, East Lansing, MI 48824, USA; lajnefni@msu.edu; 5Chongqing Key Laboratory of Nonlinear Circuits and Intelligent Information Processing, Southwest University, Chongqing 400715, China

**Keywords:** mechanical metamaterials, thermomechanical materials, self-triggered response, thermal excitations

## Abstract

In this study, we propose self-triggered thermomechanical metamaterials (ST-MM) by applying thermomechanical materials in mechanical metamaterials designed with asymmetric structures (i.e., microstructural hexagons and chiral legs). The thermomechanical metamaterials are observed with programmable mechanical response under thermal excitations, which are used in mechanical metamaterials to obtain chiral tubes with negative Poisson’s ratio and microgrippers with temperature-induced grabbing response. Theoretical and numerical models are developed to analyze the thermomechanical response of the ST-MM from the material and structural perspectives. Finally, we envision advanced applications of the ST-MM as chiral stents and thermoresponsive microgrippers with maximum grabbing force of approximately 101.7 N. The emerging ST-MM provide a promising direction for the design and perception of smart mechanical metamaterials.

## 1. Introduction

Mechanical metamaterials (MM) have expanded the horizon of programming mechanical characteristics for bulk materials from the structural perspective. The superior properties of metamaterials have also attracted extensive research interests in other areas such as acoustics metamaterials [1,2], optical metamaterials [3], thermal metamaterials [4], etc. Typically, mechanical performance programmability of MM is based on geometric nonlinearity, i.e., the unique mechanical characteristics are due to the microstructures while the materials remain in the linear elastic domain to avoid permanent damages [5,6]. For example, lattice and auxetic microstructures are designed to obtain negative Poisson’s ratio, origami and kirigami approaches are applied for self-folding metamaterials [7], corrugated microstructures are assembled in plate-like structures to enhance bending stiffness [8], etc. Programmable mechanical response has been reported on MM, including complete recovery from large strains in compression [9,10,11], tension [12], and twisting [13]; ultralightness and ultrastiffness [14,15], the tunable Poisson’s ratio [16]. More recently, research interests have been shifted to tailor the mechanical performance of MM from the material perspective by using artificial materials [17]. For example, functionally graded materials, carbon nanotubes, and graphene reinforcement have been applied in MM [17,18]. Structurally and materially tuning MM has led to creative applications in various fields, e.g., mechanical sensors [19], temperature controllers [20], shock and sound absorbers [21,22], artificial muscles [23], biosensors [24], and impenetrable bulletproof devices [25]. The main research effort has been dedicated to the structural design and optimization of MM at the current stage, while only a few studies have been carried out to exploit the material properties of MM using advanced materials such as thermomechanical materials.

Thermomechanical materials have been developed as the type of self-adaptive materials triggered by external temperature. Studies have been conducted to investigate the material properties of thermomechanical materials. Experimental fabrication and characterization were carried out on the microstructures of the thermomechanical materials, e.g., cellular materials [26], 2D materials [27], and vacuum-infused thermoplastic- and thermoset-based composites [28]. Besides the experimental characterization, constitutive models have been developed to predict the thermomechanical response [29,30]. For example, theoretical studies were conducted to predict the thermally-responsive behavior of the granular materials [31,32], the elastic-viscoplastic response of amorphous thermoplastic polymers [33], the phase transitions of shape memory polymers [34], and the constitutive response of the crystallizable triple shape memory polymers [35,36]. Thermomechanical materials have been incorporated in various advanced devices, such as endoscopic surgery suture, orthopedic cast, aneurysm occlusion devices, cell proliferation study, and antitumor effects [37]. Given the self-adaptive response subjected to temperature, thermomechanical materials can be used in MM for more programmable mechanical response and multifunctional applications.

In this study, we develop self-triggered thermomechanical metamaterials (ST-TM) by incorporating thermomechanical materials into asymmetric MM (i.e., chiral tubes and origami microgrippers). The ST-TM are subjected to the external temperature and the self-adaption is investigated with respect to the asymmetric structures. The material and structural properties of the ST-TM are theoretically characterized, and then the mechanical response is numerically studied with respect to the negative Poisson’s ratio of the chiral tubes and the grabbing configuration of the origami microgrippers. The theoretical and numerical are compared with satisfactory agreements. In the end, we envision the advanced applications of the ST-TM for chiral stents and thermoresponsive microgrippers. The emerging ST-TM provide a new version in design, manufacture and perception of smart MM.

## 2. Principles, Materials and Methods

To achieve the self-trigger response of the ST-TM in the temperature field, thermomechanical materials are designed as chiral and origami metamaterial structures. In this section, two types of editable ST-TM (i.e., chiral tubes and origami microgrippers) are characterized from the structural and material perspectives.

### 2.1. Design Principles of the Asymmetric ST-TM

Comparing to the symmetric structures, the ST-TM are observed with more programmable performance due to the asymmetry from the chiral and origami-inspired, layered structures. The thermomechanical material considered is polyethylene (PE), with density of 1.25 g/cm^3^, Young’s modulus of 2250 MPa, Poisson’s Ratio of 0.3, and thermal expansion coefficient of $\alpha =2.1\times 10^{-4}/K$.

#### 2.1.1. Chiral Tubes

Figure 1a presents the structure of a chiral tube composed of the hexagonal cells and chiral legs. Chiral structures are typically characterized as asymmetric structures that show the feature of mirror symmetry. The use of ligament bending and deformation characteristics and node rotation of chiral elements has led to the proposal of various two-dimensional and three-dimensional chiral mechanical metamaterials [37]. When the two ends of the chiral tube are subjected to a force F, the hexagonal cells rotate and the chiral legs are elongated, which is shown in Figure 1b. When the chiral tube reaches the equilibrium state, the rotation angle of the cells is θ. The forces from the chiral legs in the six directions are, $F_{1}^{'}$, $F_{2}^{'}$, $F_{3}^{'}$, $F_{4}^{'}$, $F_{5}^{'}$ and $F_{6}^{'}$. The relations between these forces are:(1)$$\left\{\begin{array}{c} F_{1}^{'}=F_{4}^{'}\\ F_{2}^{'}=F_{5}^{'}\\ F_{3}^{'}=F_{6}^{'} \end{array}\right.$$

For each force, the angle between the equilibrium direction and the initial direction is $\theta^{\prime }=\frac{\pi }{6}-\theta$. The change in the magnitude of each force during the rotation is regarded as linear. Therefore, when the rotation angle of the force is β, the total bending moment produced by forces in six directions can be written as:(2)$$M_{Toatal,\;\beta }=\frac{2\beta L\sin\left(\frac{\pi }{6}-\theta -\beta \right)\left(F_{1}^{'}+F_{2}^{'}+F_{3}^{'}\right)}{\frac{\pi }{6}-\theta },$$
where L is the side length of the hexagonal cells. The average total bending moment is:(3)$$\bar{M}=\frac{\int_{0}^{\theta^{\prime }}M_{Total,\;\beta }d\beta }{\theta^{\prime }}=\frac{2L\left(F_{1}^{'}+F_{2}^{'}+F_{3}^{'}\right)\left[\left(\frac{\pi }{6}-\theta \right)-\sin\left(\frac{\pi }{6}-\theta \right)\right]}{{\left(\frac{\pi }{6}-\theta \right)}^{2}}$$

The rotation of the hexagonal cells can be determined as:(4)$$\theta =\frac{\bar{M}\delta }{G_{0}I_{p}}=\frac{16\sqrt{3}\delta \left(F_{1}^{'}+F_{2}^{'}+F_{3}^{'}\right)\left[\left(\frac{\pi }{6}-\theta \right)-\sin\left(\frac{\pi }{6}-\theta \right)\right]}{15G_{0}L^{3}{\left(\frac{\pi }{6}-\theta \right)}^{2}}$$
where δ is the thickness of the chiral tube, $I_{p}$ is the polar moment of inertia of the hexagonal cells, $G_{0}$ is the shear modulus of the material. Based on the force balance, we have:(5)$$\left\{\begin{array}{c} F=\left(N_{x}+1\right)F_{1}^{'}+\frac{N_{x}}{2}\left[F_{2}^{'}\sin\left(\frac{\pi }{3}-\theta \right)+F_{3}^{'}\sin\theta \right]\\ F_{2}^{'}=F_{3}^{'}=F_{1}^{'}\cos60^{{}^{\circ}} \end{array},\right.$$
where $N_{x}$ and $N_{y}$ are the number of the triangles in the x and y directions, respectively. Solving Equation (5), we have:(6)$$\left\{\begin{array}{c} F_{1}^{'}=\frac{4F}{4\left(N_{x}+1\right)+N_{x}\left[\sin\left(\frac{\pi }{3}-\theta \right)+\sin\theta \right]}\\ F_{2}^{'}=F_{3}^{'}=\frac{2F}{4\left(N_{x}+1\right)+N_{x}\left[\sin\left(\frac{\pi }{3}-\theta \right)+\sin\theta \right]} \end{array}\right.$$

Therefore, the rotation angle is:(7)$$\theta =\frac{128\sqrt{3}F\delta \left[\left(\frac{\pi }{6}-\theta \right)-\sin\left(\frac{\pi }{6}-\theta \right)\right]}{15G_{0}L^{3}{\left(\frac{\pi }{6}-\theta \right)}^{2}\left\{4\left(N_{x}+1\right)+N_{x}\left[\sin\left(\frac{\pi }{3}-\theta \right)+\sin\theta \right]\right\}}$$

The deformation of the chiral legs includes three parts, namely the deformation of area 1 and area 3 and the bending of area 2. It can be written as:(8)$$\Delta_{i}=\delta_{i1}+\delta_{i2}+\delta_{i3},$$
where $\Delta_{i}$ is the total deformation when the chiral legs are pulled by $F_{i}$. Note that $\delta_{i1}$, $\delta_{i2}$ and $\delta_{i3}$ are the deformation of area 1, area 2 and area 3. The calculation formula of $\delta_{i1}$, $\delta_{i2}$, $\delta_{i3}$ is:(9)$$\left\{\begin{array}{c} \delta_{i1}=\delta_{i3}=\frac{F_{i}\left(g-t\right)}{2E_{0}t\delta }\\ \delta_{i2}=\frac{4F_{i}L^{3}}{E_{0}\delta t^{3}} \end{array}\right.,$$
where g is the distance between two cells, t is width of chiral legs, $E_{0}$ is the Young’s modulus of the material. $\Delta_{i}$ can be written as:(10)$$\Delta_{i}=\frac{F_{i}\left(gt^{2}-t^{3}+4L^{3}\right)}{E_{0}\delta t^{3}}$$

The elongation of chiral legs can be obtained with respect to the direction of the force changes $\theta^{\prime }$ as:(11)$$\Delta_{i}^{'}=\Delta_{i}\cos\theta^{\prime }=\frac{F_{i}\left(gt^{2}-t^{3}+4L^{3}\right)\cos\left(\frac{\pi }{6}-\theta \right)}{E_{0}\delta t^{3}}$$

There are currently two main ways to design chiral tubes. The first is to increase the thickness of the chiral plate to be much larger than the scale of chiral elements, such as chiral tubes based on hierarchical anti-tetrachiral metastructures and hierarchical anti-hexachiral metastructures [38]. This design has increased the energy absorption efficiency and compression resistance. The second is to curl the chiral plate in the axial direction to form a cylinder, such as the novel tetrachiral architected cylindrical tubes [39]. This design has more obvious axial torsion, bending or expansion and contraction performance. Here, we choose the second design. The design parameters of the chiral tube are shown in Table 1, and the chiral tube model is presented in Figure 1a.

Therefore, the elongations of radial height and axial circumference of chiral tubes are:(12)$$\left\{\begin{array}{c} \Delta H=\frac{(N_{r}-1)F_{1}^{'}\left(gt^{2}-t^{3}+4L^{3}\right)\cos\left(\frac{\pi }{6}-\theta \right)}{E_{0}\delta t^{3}}\\ \Delta C=\frac{N_{a}\left(F_{2}^{'}+F_{3}^{'}\right)\left(gt^{2}-t^{3}+4L^{3}\right)\cos\left(\frac{\pi }{6}-\theta \right)\cos30^{{}^{\circ}}}{2E_{0}\delta t^{3}} \end{array}\right.,$$
where $N_{r}$ and $N_{a}$ are the number of hexagonal cells in the radial and axial directions, respectively. Taking Equation (12) into Equation (6), the increment of the axial circumference of the chiral tube under radial tension yields:(13)$$\Delta C=\frac{N_{a}W}{J\left(\theta \right)},$$
where $W=\frac{15G_{0}L^{3}\left(gt^{2}-t^{3}+4L^{3}\right)}{128E_{0}\delta^{2}t^{3}}$ and $J\left(\theta \right)=\frac{\left(\frac{\pi }{6}-\theta \right)-\sin\left(\frac{\pi }{6}-\theta \right)}{\begin{array}{c} {\left(\frac{\pi }{6}-\theta \right)}^{2}\theta \cos\left(\frac{\pi }{6}-\theta \right) \end{array}}$.

Therefore, the increment of the chiral tube’s diameter is:(14)$$\Delta d=\frac{\Delta C}{\pi }=\frac{N_{a}W}{\pi J\left(\theta \right)},$$
and the Poisson’s ratio of this tube is:(15)$$\nu =-\frac{\Delta C}{\Delta H}\frac{H}{C},$$
where H and C are the initial radial height and axial circumference of chiral tube, respectively.

The thermal expansion-induced displacements of the chiral tube in the axial and radial directions due to the temperature change ΔT are:(16)$$\left\{\begin{array}{c} \Delta C_{e}=\frac{1}{2}C\alpha \Delta T\\ \Delta H_{e}=\frac{1}{2}H\alpha \Delta T \end{array},\;\right.$$
where α is thermal expansion coefficient considering the structure and material properties. When the chiral tube is fixed in the radial direction, the displacement in the axial direction can be expressed as:(17)$$\Delta D=\frac{\alpha C\Delta T\left(1+\nu \right)}{\pi }$$

#### 2.1.2. Origami Microgrippers

The origami microgrippers are inspired by the principle of rigid origami, where the structures are deformed due to the bending of the origami-enabled creases to achieve folding. There have been many methods to obtain grippers, such as using elastomeric polymer that responds to an external light [40,41,42]. Thermomechanical materials are applied here. In particular, the creases are designed with thermomechanical materials, and the origami planes are designed with rigid materials. And the overlapping area of the crease and the two adjacent origami planes is not equal, thus reflecting the asymmetric feature. The folding process of the origami metamaterial model is shown in Figure 2. The thermomechanical materials are expanded under thermal excitations, such that the entire origami microgrippers are triggered to grab like the human hand. The folding angle of the origami microgrippers is tuned by changing the structural design parameters at the creases, as detailed in Figure 2b.

According to the deformation scenarios of the microgripper in Figure 2c, the relationship between the folding angle and the design parameters can be written as:(18)$$\Delta L=\frac{\left(W_{\mathrm{gap}}-W_{\mathrm{bridge}}\right)\alpha \Delta T}{2}$$
and
(19)$$\sin\frac{\theta }{2}=\frac{\Delta L}{2T_{\mathrm{expansion}}}$$
where θ is the single-sided folding angle, ΔL is the thermal deformation, $T_{\mathrm{expansion}}$ is the thickness of the elastic material, $W_{\mathrm{gap}}\;$and $W_{\mathrm{bridge}}$ are the widths of the upper and lower surfaces of the creases, respectively, β=2θ is the final folding angle.

Figure 2c demonstrates the origami-inspired microgripper composed of four fingers, each finger having five thermally expansible joints. The relationship between the folding angle and the structural design parameters is obtained when the thickness of the crease is constant, the folding angle is increased as the crease width is increased. However, when the crease width is constant, the folding angle is decreased as the crease thickness is increased. Figure 2d displays the force analysis when the gripper grasps the object. The force along z-direction is balanced. The force along the last joint is written as:(20)$$F=\frac{1}{2}E\alpha \Delta TS$$
and
(21)$$F_{1}=Fsin\left(\theta_{5}-\theta_{4}\right)$$

The balanced equation along z-direction can be written as:(22)$$F_{1}\sin\left(\frac{2\pi }{3}-\theta_{5}\right)=\frac{1}{4}G$$
where $\theta_{4}$ is the folding angle when the fourth joint is triggered, $\theta_{5}$ is the folding angle when the fifth joint is triggered. G is the gravity of the object that grippers grasp.

### 2.2. Numerical Simulations of the Asymmetric ST-TM

The proposed chiral tubes and microgrippers are numerically simulated in Abaqus, particularly using the dynamic/explicit solving algorithm for its efficiency. The rigid parts of the microgrippers are defined as the cast steel with density of $\rho =1.25\;g/{\mathrm{cm}}^{3}$, Young’s modulus of E=176,000 MPa, and Poisson’s ratio of v=0.3. Figure 3 presents the numerical models of the chiral tubes and origami microgrippers. The geometric and material properties are in Table 2. In the chiral tube model, the parameters of thermally expansive materials mentioned in Section 2.1 are applied to the chiral tube, and the upper part of the tube is fixed. The model was separated into 78,899 elements with Tet mesh type. To carry out the deformation of the chiral tubes, the temperature range of 0–5 ℃ is evenly applied to the structures within the time period of 10 s during the analytical step. In the microgrippers model, the parameters of thermally expansive materials and rigid materials are applied to the joints and the other parts, respectively. The palm part of the tube is fixed, and the whole parts were separated into 25,600 elements with Hex mesh type. To carry out the midspan expansion of the microgrippers, the temperature range of 0–70 ℃ is evenly applied to the structures within the time period of 600 s during the analytical step, which is applied to each joint consequently.

## 3. Results and Discussion

### 3.1. Results for the Chiral Tubes

The whole heating process of the chiral tubes is demonstrated in Figure 4. On account of the low-temperature interval, the stress triggered by temperature is limited; however, the deformation is distinct. Figure 5a displays the relationship between chiral ratio, axial stretching, and rotation angle. The dash part is the region where the rotation angle is more than 0.1° so that the axial stretching here is significant. Figure 5b compares the axial elongation and rotation angle between the chiral and regular tubes. It can be seen that the axial elongation of the chiral tube decreases when λ is gradually increased, which goes to show that if λ is further increased, the axial elongation approaches 0. The axial elongation of the chiral tubes increases from 0.183 to 0.789 mm with the increase of λ. Therefore, the axial elongation of the chiral tubes can be controlled by tuning the chiral metamaterials while maintaining the overall size and material properties the same. On the other hand, the torsion angle of the chiral tubes increases when λ is gradually decreased. Since the temperature increment is set to 0–5 ℃ in this study, the torsion angle is relatively small. When λ is reduced from 0.82 to 0.69, the torsion angle reduces by 78%, which can be explained by the chiral tubes twisting more significantly when the temperature is increased faster. More importantly, when the chiral ratio ranges from about 0.69 to 0.71, the rotation angle and the axial stretching are both large. As a consequence, this region can be utilized to obtain more sensitive self-triggered ST-TM. Table 3 presents the numerical results of the axial elongation and rotation angle for the chiral and regular tubes.

There exist several limitations in the FE models described here. To improve the computational quality in simulation, meshes with small scales are always generated; however, computing with many small cell units in complicated metamaterials structures is time-consuming. The chiral tube we simulated here is small, thus, the regular method is sufficient. In large-scale engineering applications, more hexagonal cells or larger dimensions must be used. The mechanical properties of heterogeneous materials such as the chiral metamaterials explored here also have uncertainty at both the macro- and microscales. Hence, some approaches like computational homogenization [43], which can account for complicated and possibly nonlinear structural behavior, are used to simulate metamaterial response in large-scale models.

### 3.2. Results for the Origami Microgrippers

Folding deformation is numerically and theoretically studied for the origami microgrippers. Figure 6 presents the variation of the folding angles for the grippers with different crease widths, thicknesses, and numbers. When the crease thickness is constant, the folding angle is increased as the crease width is increased, as shown in Figure 6a. When the crease width is constant, on the other hand, the folding angle is decreased with the crease thickness, as shown in Figure 6b. Comparing the numerical and theoretical results, the difference is more significant when the crease width is constant, which is less crucial when the crease thickness is constant. This is because the thickness influence of the rigid material is considered on the folding angle of the microgrippers in the theoretical model. Therefore, when the crease thickness is critically different from the thickness of the rigid plane, the theoretical angle is significantly different from the actual one. The maximum difference is up to 42.7%. In addition, the folding angle of the microgrippers is increased with the number of the creases, and the linear relationship is obtained, as shown in Figure 6c.

In this study, two kinds of heating strategies are applied to the ST-TM: 1) Segmented heating, which refers to the fact that the creases are heated to 70 ℃ in sequence; and 2) global heating, which means the entire origami microgrippers are heated at the same time. The folding angles based on both of the heating strategies are the same, which indicates that the folding of different creases is independent. The microgrippers can be heated by either of the methods to obtain the desired folding angle. The theoretical and numerical models accurately predict the changes of the folding angle with the number of the creases, as shown in Figure 6d. The crease angle variation cases considered in this study include the thickness varied from 0.25, 0.5, 0.75 to 1 mm, the width from 4, 6, 8, 10, 12, 14 to 16 mm, and the folding number from 1, 2, 3, 4, 5 to 6. Table 4 presents the folding angles of the origami microgrippers with respect to the structural design parameters (i.e., the crease width, the crease thickness, the number of creases).

Figure 7 indicates the folding configurations of the origami microgrippers with time. It can be seen that the inner stress is mainly concentrated in the creases with thermal expansion. Therefore, the folding of the microgrippers is obtained by the thermal-induced deformation of the creases. On the contrary, the rigid plane only has the rigid rotations generated by the creases, so the stress is relatively small and evenly distributed. According to the numerical results, the microgrippers realize multiangle self-folding under external thermal changes, besides, according to the Equations (20)–(22) and the numerical results of folding angle in Figure 6d, $\theta_{4}=73.93{}^{\circ}$, $\theta_{5}=62.2{}^{\circ}$. Therefore, the maximum force (i.e., the maximum gravity G of the object that the gripper can grasp) supplied by the grippers is about 101.7 N.

### 3.3. Vision for Potential Applications

#### 3.3.1. Potential Applications as Chiral Stents

Figure 8 presents the potential applications of chiral tubes as chiral stents and origami grippers as thermoresponsive microgrippers. Chiral tubes with both thermomechanical material properties and chiral structural properties can be used as chiral stents. The thermomechanical material ensures that the stents can be triggered by temperature, which leads to the axial expansion when the temperature rises. Under axial deformation, chiral structures results in negative Poisson’s ratio, which amplifies the axial deformation of stents. The chiral structures have been reported with promising mechanical properties such as enhanced shear modulus, indentation resistance, and fracture toughness [7], which ensure that the stents have good stability. Chiral tubes can be designed at the millimeter scale and embedded in the human body (e.g., blood vessels and bronchi). Human body temperature is an important indicator reflecting human health. A specific human body temperature will trigger the stent to produce a specific axial expansion deformation. By using piezoelectric materials or triboelectric materials to capture this deformation coupled with human body temperature data, the effect of diagnosing human health can be achieved. In addition, the stents can realize self-adaptability by accurately controlling its mechanical deformation (i.e., negative Poisson’s ratio) in the temperature field. It can be used to support contracted blood vessels and maintain the smooth transportation of human blood. Complex program designs and high-cost technical support results in inefficient applications for some existing systems such as medical chat robots [44,45]. Therefore, chiral stents based on chiral tubes are expected to be used in the biomedical field as a competitive alternative.

#### 3.3.2. Potential Applications as Thermoresponsive Microgrippers

Based on the folding response of the ST-TM, the thermoresponsive microgrippers are envisioned. The origami microgrippers exhibit the programmable folding angles under specific working conditions, i.e., the size or weight of the holdable objects can be determined using geometric and material properties. Although the working conditions are selected with constant temperature changes, the holding performance of the thermoresponsive microgrippers can be designed with the geometries and materials. Therefore, the thermoresponsive microgrippers can be programmed to achieve the gripping and releasing response on different objects. This function is expected to provide applications in, for example, quality monitoring of particle size. Traditional manipulators are typically driven by a mechanical force or electrical signal and need to have peripheral controllers to operate, which critically affects the flexibility of those devices. The reported thermoresponsive microgrippers are envisioned to break these limitations and operate by temperature. At present, many temperature control technologies and strategies have been proposed to control room temperature and microscopic temperature, which can create a suitable working environment for microgrippers. For example, organic electronic devices based on organic thermoelectric materials [46], a modified active disturbance rejection control (ADRC) [47] and temperature-based microscopic temperature control of sensitive upconversion nanocomposite [48]. Besides, the single-chip microcomputer is also widely used as an advanced temperature control method [49].

## 4. Conclusions

In this study, we developed asymmetric self-triggered thermomechanical metamaterials (ST-TM) for programmable mechanical response subjected to external thermal fields. The ST-TM were obtained by incorporating the thermomechanical materials into MM with chiral and origami-inspired structures. Numerical models were developed to analyze the negative Poisson’s ratio for the chiral tubes and the temperature-induced grabbing response for the origami microgrippers. We investigated the ST-TM with different thermal expansion properties and the corresponding mechanical response were observed and discussed. The parametric studies indicated the programmability of the ST-TM in the temperature field by changing material properties. It is worth mentioning that the complex-valued, intensity-induced losses are important to FE models; we need to consider this in future work. In the end, the ST-TM are envisioned for advanced applications in the chiral stents and thermoresponsive microgrippers. The emerging ST-TM provides a promising direction for the smart MM enabled by functional materials.

## Figures and Tables

**Figure 1 materials-14-02177-f001:**
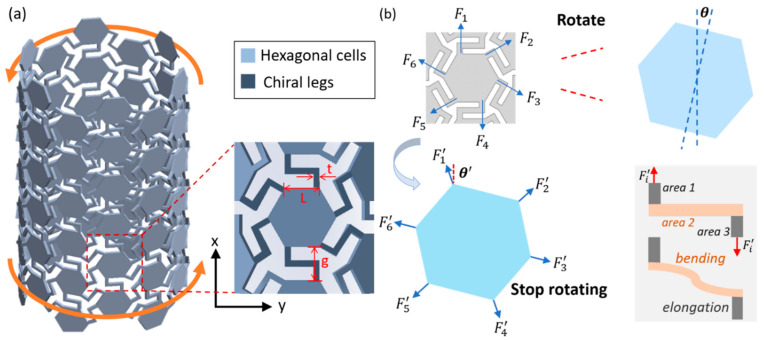
(**a**) Chiral tube assembled by hexagonal units and chiral legs. (**b**) Deformation of hexagonal cells and chiral legs.

**Figure 2 materials-14-02177-f002:**
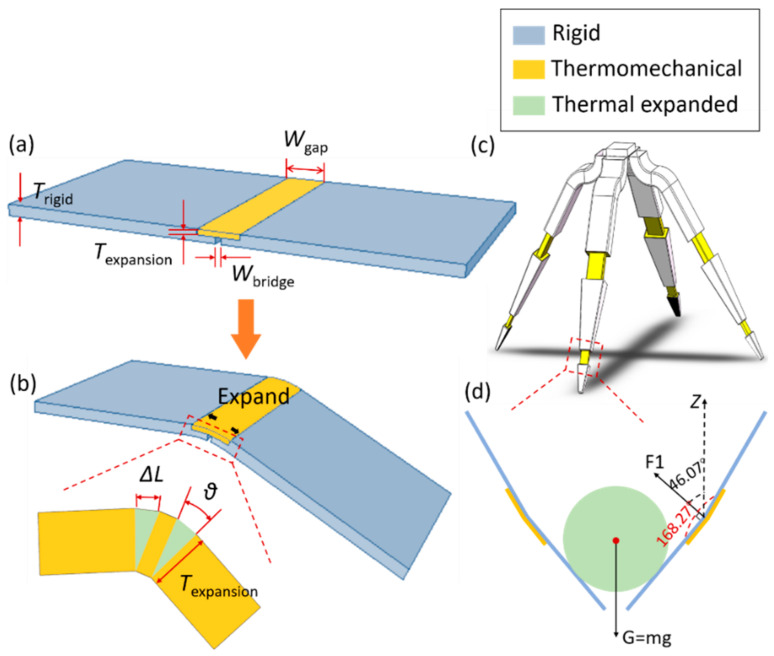
Folding process (**a**) before being heated (unfolded); (**b**) after being heated (folded); (**c**) Self-folding microgripper based on the origami metamaterials; (**d**) force analysis when holding the object.

**Figure 3 materials-14-02177-f003:**
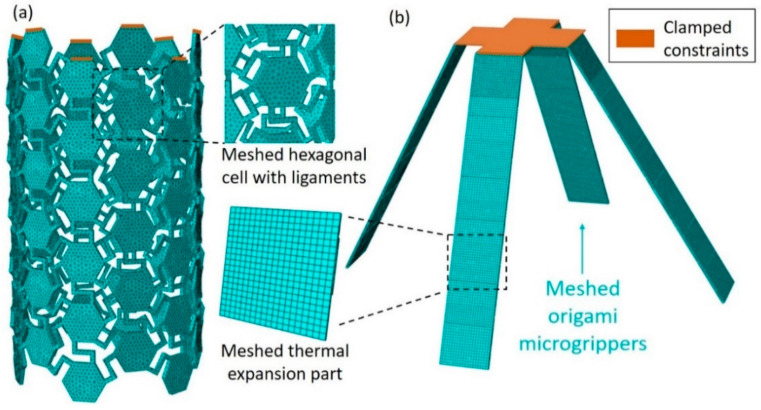
Numerical models of the (**a**) chiral tube subjected to the temperature range of 0–5 ℃ and (**b**) origami microgripper under the temperature range of 0–70 ℃.

**Figure 4 materials-14-02177-f004:**
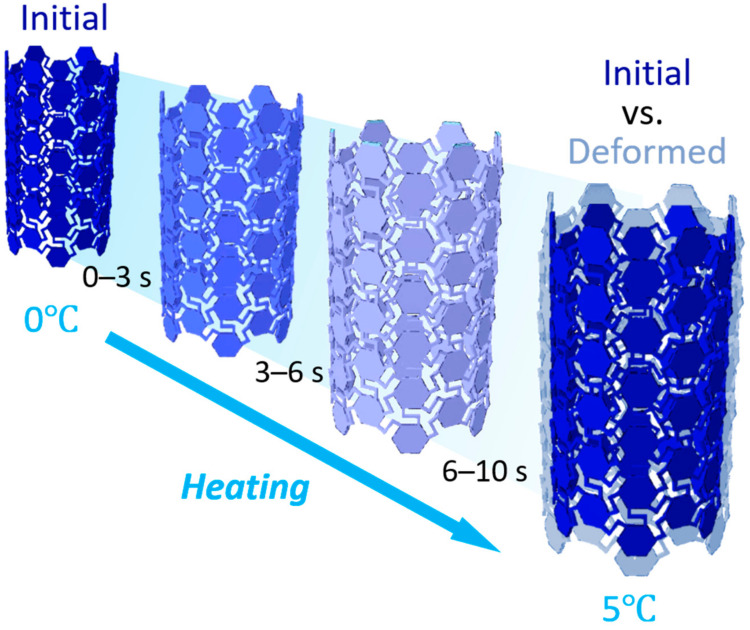
Thermal expansion process of the chiral tubes triggered by temperature changes.

**Figure 5 materials-14-02177-f005:**
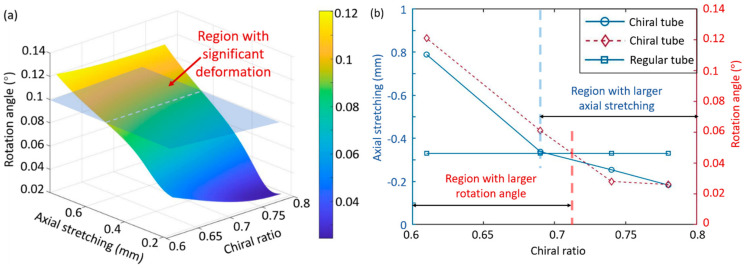
(**a**) The axial stretching and rotation angle vary with the chiral ratio. (**b**) Comparison between the chiral tube and the regular tube.

**Figure 6 materials-14-02177-f006:**
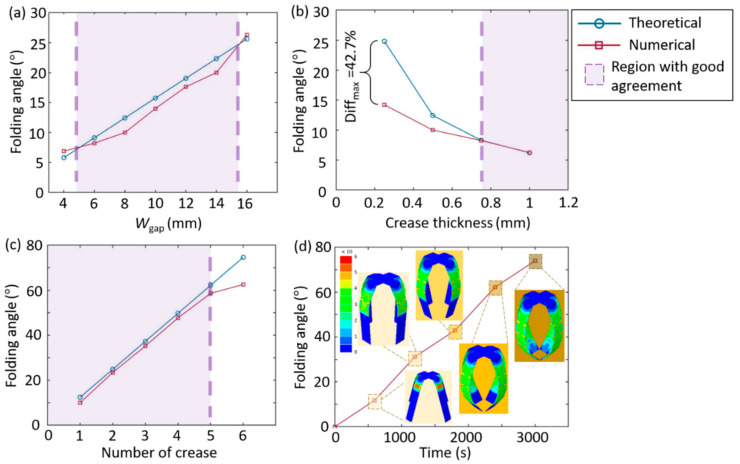
The crease angle variations of the microgrippers with respect to the (**a**) crease width; (**b**) crease thickness and (**c**) number of creases. (**d**) Variation of the entire folding angle with heating time for the microgrippers.

**Figure 7 materials-14-02177-f007:**
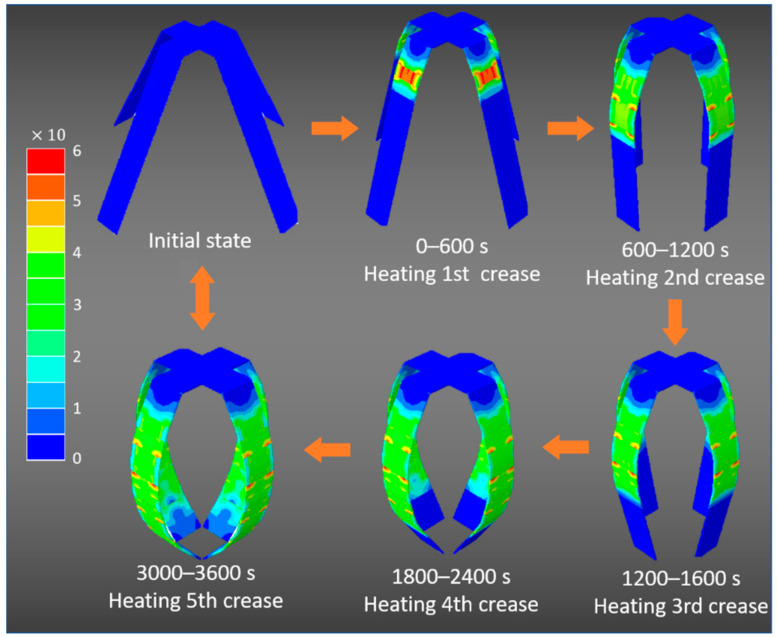
Self-folding configurations with time for the origami microgrippers triggered by temperature changes.

**Figure 8 materials-14-02177-f008:**
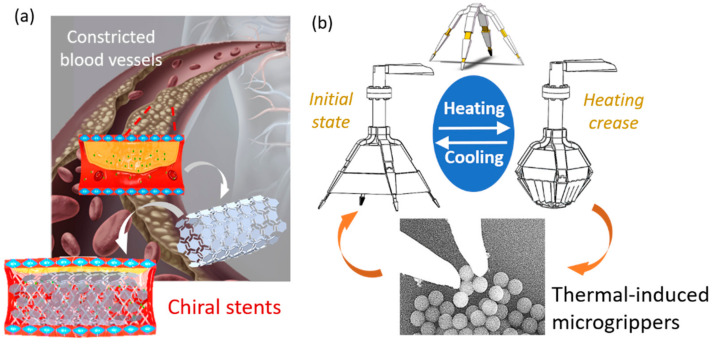
Potential applications of (**a**) chiral stents and (**b**) origami grippers as thermoresponsive microgrippers.

**Table 1 materials-14-02177-t001:** Design parameters of the chiral tube.

Fixed factors (mm)		Variables (mm)		Chirality Ratio λ
Length of hexagons L	5.77	Thickness t	0.50 0.75 1.00 1.25 1.5	0.87 0.82 0.78 0.74 0.69
The gap of hexagons g	5.5			

**Table 2 materials-14-02177-t002:** Geometric and material properties and loading conditions of the chiral tube and origami microgrippers.

Material Properties			Geometric Properties					Loading
Rigid materials	Density (kg/mm3)	1.25$\times 10^{-6}$	Chiral tube	Overall (mm)		Chirality Ratio λ		0–5 ℃
	Young’s modulus (MPa)	176,000		Length	180.5	0.78		
						0.74		
	Poisson’s Ratio	0.3		Diameter	50	0.69		
						0.61		
Thermally expansive materials	Density (kg/mm^3^)	1.25$\times 10^{-6}$	Origami microgrippers	Height	112	Crease geometries (mm)		0–70 ℃
	Young’s modulus (MPa)	2250		Width	180	$T_{e}$	0.5	
	Poisson’s Ratio	0.3		Angle between fingers and plate (°)	120	$W_{\mathrm{gap}}$	8	
	Expansion coefficient	$2.1\times 10^{-4}$					16	

**Table 3 materials-14-02177-t003:** The elongation and rotation angle of the chiral and regular tubes.

Type	Length (mm)	Diameter (mm)	Chirality Ratio λ	Axial Stretching (mm)	Rotation Angle (°)
Chiral tube	180.5	50	0.78 0.74 0.69 0.61	0.183 0.254 0.338 0.789	0.026 0.028 0.061 0.121
Regular tube			—	0.330	0.000

**Table 4 materials-14-02177-t004:** The crease angle varies with the crease width, crease thickness, and the number of creases.

Folding Angle (°)					
Case 1 (Crease Thickness = 0.5)		Case 2 (Crease Width = 8)		Case 3 (Thickness = 0.5, Width = 8)	
Theoretical	Numerical	Theoretical	Numerical	Theoretical	Numerical
5.81	6.90	24.8	14.21	12.44	10.00
9.13	8.23	12.44	10.00	24.88	23.47
12.44	10.00	8.30	8.26	37.32	35.22
15.75	14.00	6.22	6.25	49.76	47.72
19.05	17.63	—	—	62.20	58.63
22.34	20.00			74.64	62.54
25.62	26.30			—	—

## Data Availability

The data presented in this study are available on request from the corresponding author after obtaining permission of authorized person.

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
