# Peer review of "Self-Triggered Thermomechanical Metamaterials with Asymmetric Structures for Programmable Response under Thermal Excitations"

_materials, 2021, doi:10.3390/ma14092177_

Round 1

Reviewer 1 Report

Dear authors,

the study can potentially be interesting for the readers. Nevertheless, it suffers from several major issues in introduction, design and discussion. It must be greatly improved before any publishing.

Abstract

L11: It is not clear whether this sentence stated what is generally done or it is a result fo this study. Perhaps, avoiding the first sentence would help to increase the readability.

L14-L16: These sentences are unclear and perhaps redundant. It requires polishing.

Introduction

L31: It is not clear what is meant by this sentence: The MM can have superior mechanical properties achieved by carefully designing the geometry and topology of the underlying microstructure.

L31: Formatting inconsistency

L32: lattice and auxetic cannot be simply grouped (for example, lattice can have auxetic properties): the lattice is a structure and auxeticity is rather a property. Please reformulate

L43: This is confusing, as controlling the MM by external physical fields is of active development, please provide references for those with thermal field and discuss concerning your design.

L61: Using „self-triggered“ must be explained in the context of metamaterial as it is taken from other scientific fields. It is not clear what the authors mean by using this word combination.

Materials and Methods

L73: It is not clear how the thermally sensitive material can be designed in metamaterial – It could rather be a combination of structural design from a bulk material that is thermally sensitive.

L79: It is known that symmetry and periodicity constraint the design space of metamaterial, it must be explained to the reader where the asymmetry allows more control over the design in this study. If it is a novel way how to use asymmetry it must be clearly explained or cited if this has been already explored in the literature.

L80: Describing the material only with some engineering constant is not sufficient, what kind of material was used?

L83: Is the chiral tube design novel? If not the novel, please provide an original reference.

L86: The coordinate system is missing, it is not clear how to consider cartesian coordinates for a structure, which is cylindrical and hence radial versus axial deformation would better be suited.

L90: How was Poisson’s ratio expression (2) derived?

L100: It is not clear, what bulk material was used, the thermomechanical one described above?

L102: Fixing the ends and thermally loading the tube will not lead to an axial displacement, but rather to some radial buckling or another deformation (and the expression 4 does not hold). What is meant by this formulation?

L134: It is clear how the thermally sensitive material is used, but it has little to do with origami. Origami is rather a design principle.

L136: Reasons for employing a numerical model must be clearly explained. What was achieved by numerical models? Why was an explicit solver required?

L137: It is highly confusing to call the gripper origami-inspired. Please consider removing the similarity with origami.

L140: What is the material used for rigid parts of the gripper? It is not sufficient only describe it by some constants.

L141: To be able to replicate your results and validate, one must properly define why the hexahedral elements were used, how many elements were required to get a solution convergent. Moreover, using the name C3D20R without any description is highly insufficient for readers. Please provide a detailed description of the finite element model.

L142: What is meant by „clamped“ boundary conditions? Does it mean constraining all six degrees of freedom? Please reformulate.

L142: What were boundary conditions used for a thermal model?

L142: What is mean by multi-angle folding?

L151: What were the quantities computed by FE to evaluate metamaterial performance?

Results:

L222: Which stress? This is expectable as the gripper is not loaded (forces acting oppositely then folding). What were the gripper forces? How precisely can be the gripper controlled by a thermal field?

L232: What is meant by „self-diagnosis“?

L237: Using auxetic stents can be found in clinical applications. It is not clear how to design the stent in nanoscale, where the physics is very different.

Discussion and Conclussion:

This section must be greatly improved. Please compare your design of the chiral tube with other designs in the literature.

Please discuss more finely how the thermal field can be precisely controlled and inserted into a microgripper.

What precision and holder force could be achieved with this design with respect to other designs of microgrippers.

Please discuss the computational model in terms of its limitations and possible extensions.

Computing with many small cell units becomes quickly expensive and hence some approaches like computational homogenization are used to simulate metamaterial response in large scale models, please discuss this with respect to your chiral tubes and for example with the approach used here:

https://doi.org/10.1016/j.euromechsol.2019.103825

Author Response

Comment 1

Abstract

L11: It is not clear whether this sentence stated what is generally done or it is a result fo this study. Perhaps, avoiding the first sentence would help to increase the readability.

L14-L16: These sentences are unclear and perhaps redundant. It requires polishing.

Response

We thank the reviewer for the suggestion and have written the abstract in the revised manuscript. The changes are highlighted in yellow.

Comment 2

Introduction

L31: It is not clear what is meant by this sentence: The MM can have superior mechanical properties achieved by carefully designing the geometry and topology of the underlying microstructure.

L31: Formatting inconsistency

L32: lattice and auxetic cannot be simply grouped (for example, lattice can have auxetic properties): the lattice is a structure and auxeticity is rather a property. Please reformulate

L43: This is confusing, as controlling the MM by external physical fields is of active development, please provide references for those with thermal field and discuss concerning your design.

L61: Using „self-triggered“ must be explained in the context of metamaterial as it is taken from other scientific fields. It is not clear what the authors mean by using this word combination.

Response

The Introduction has been completely written in the revised manuscript. The changes are highlighted in yellow.

Comment 3

L73: It is not clear how the thermally sensitive material can be designed in metamaterial – It could rather be a combination of structural design from a bulk material that is thermally sensitive.

L79: It is known that symmetry and periodicity constraint the design space of metamaterial, it must be explained to the reader where the asymmetry allows more control over the design in this study. If it is a novel way how to use asymmetry it must be clearly explained or cited if this has been already explored in the literature.

L80: Describing the material only with some engineering constant is not sufficient, what kind of material was used?

L83: Is the chiral tube design novel? If not the novel, please provide an original reference.

L86: The coordinate system is missing, it is not clear how to consider cartesian coordinates for a structure, which is cylindrical and hence radial versus axial deformation would better be suited.

L90: How was Poisson’s ratio expression (2) derived?

L100: It is not clear, what bulk material was used, the thermomechanical one described above?

L102: Fixing the ends and thermally loading the tube will not lead to an axial displacement, but rather to some radial buckling or another deformation (and the expression 4 does not hold). What is meant by this formulation?

L134: It is clear how the thermally sensitive material is used, but it has little to do with origami. Origami is rather a design principle.

L136: Reasons for employing a numerical model must be clearly explained. What was achieved by numerical models? Why was an explicit solver required?

L137: It is highly confusing to call the gripper origami-inspired. Please consider removing the similarity with origami.

L140: What is the material used for rigid parts of the gripper? It is not sufficient only describe it by some constants.

L141: To be able to replicate your results and validate, one must properly define why the hexahedral elements were used, how many elements were required to get a solution convergent. Moreover, using the name C3D20R without any description is highly insufficient for readers. Please provide a detailed description of the finite element model.

L142: What is meant by „clamped“ boundary conditions? Does it mean constraining all six degrees of freedom? Please reformulate.

L142: What were boundary conditions used for a thermal model?

L142: What is mean by multi-angle folding?

L151: What were the quantities computed by FE to evaluate metamaterial performance?

Response

In response to the concerns of the reviewer:

  • L73 in the initial manuscript: The metamaterials structures are fabricated by the materials that are thermally sensitive.
  • L79 in the initial manuscript: The suggested explanation on the asymmetry has been added in the first paragraphs of "2.1.1. Chiral Tubes" and "2.1.2. Origami Microgrippers", respectively.
  • L80 in the initial manuscript: The materials used in this study is Polyethylene (PE).
  • L83 in the initial manuscript: To the best the authors’ knowledge, the chiral tubes consisted of hexagonal cells and chiral legs have not been reported in the literature.
  • L86 & L90 in the initial manuscript: We have added detailed derivation in chiral tubes and the x-axis and y-axis in chiral plates. In order to consider the difference between the planar and tubular structures, we have changed to the polar coordinates.
  • L100 in the initial manuscript: The previously introduced thermomechanical materials are applied to the chiral structures.
  • L102 in the initial manuscript: We agree with the reviewer that radial buckling may happen to the chiral tubes if the thickness of the tubes is much smaller than the length ( ). However, the thickness-to-length ratio of the reported chiral tubes are moderately large (i.e., the tubes are too thick to buckle under thermal load). As a consequence, the chiral tubes experience expansion or contraction in the transverse direction.
  • L134 & L137 in the initial manuscript: We agree that origami typically refers to the 3D structures that are formed by folding 2D plates. In fact, the thermomechanical materials are used to fold the joints of the grippers in this study, which is inspired by the origami structures. Similar ideas have been reported in the literature such as:

Tolley, M.T., Felton, S.M., Miyashita, S., Aukes, D., Rus, D., Wood, R.J. Self-folding origami: Shape memory composites activated by uniform heating. Smart Materials & Structures 23.9 (2014): 094006.

  • L136 in the initial manuscript: The thermomechanical response reported in “Results and Discussion” is obtained using the numerical model. We have compared the results obtained using the explicit and implicit solvers in Abaqus. Although, the difference between the explicit and implicit results is negligible, it is much more efficient to solve the explicit model. As a consequence, the explicit solver is used in this study.
  • L140 in the initial manuscript: In the grippers, the rigid parts are designed with rigid material such as steel. The material properties are provided in the revised manuscript.
  • L141 in the initial manuscript: The detailed description of the numerical model has been added to “2.2 Numerical Simulation of the Asymmetric ST-MM” in the revised manuscript.
  • L142 in the initial manuscript: The “clamped boundary conditions” refers to the situation that the entire six degrees of freedom are fixed.
  • L142 in the initial manuscript: For the chiral tubes, one end of is fixed while the other end is free. For the microgrippers, the end is fixed and the thermomechanical materials are free to deform such that the microgrippers can be triggered to grab.
  • L142 in the initial manuscript: Multi-angle folding is used to describe the deformation process of the microgrippers since they contain several joints that are able to fold one by one.
  • L151 in the initial manuscript: The necessary quantities are provided in the revised manuscript.

The changes have been added to the revised manuscript. The changes are highlighted in yellow.

Comment 4

L222: Which stress? This is expectable as the gripper is not loaded (forces acting oppositely then folding). What were the gripper forces? How precisely can be the gripper controlled by a thermal field?

L232: What is meant by „self-diagnosis“?

L237: Using auxetic stents can be found in clinical applications. It is not clear how to design the stent in nanoscale, where the physics is very different.

Response

In response to the concerns of the reviewer:

  • L222 in the initial manuscript: It is the inner stress of the grippers caused by the thermal load. The gripper force is determined as 101.7 N. Since the grippers are designed with several thermomechanical joints, the grabbing response can be tuned in the thermal field. To increase the accuracy of the gripper, it is necessary to include more thermomechanical joint.
  • L232 in the initial manuscript: We agree with the reviewer that it is confusing and have removed self-diagnosis in the revised manuscript.
  • L237 in the initial manuscript: In this study, the chiral stents are designed at the millimeter scale. We have changed "3.3.1. Potential Applications as Chiral Stents" the revised manuscript accordingly.

Comment 5

  1. This section must be greatly improved. Please compare your design of the chiral tube with other designs in the literature.
  2. Please discuss more finely how the thermal field can be precisely controlled and inserted into a microgripper.
  3. What precision and holder force could be achieved with this design with respect to other designs of microgrippers.
  4. Please discuss the computational model in terms of its limitations and possible extensions.
  5. Computing with many small cell units becomes quickly expensive and hence some approaches like computational homogenization are used to simulate metamaterial response in large scale models, please discuss this with respect to your chiral tubes and for example with the approach used here: https://doi.org/10.1016/j.euromechsol.2019.103825

Response

In response to the concerns of the reviewer:

  1. The suggested comparison and discussion have been added between Eqs. (20) and (21) in the revised manuscript.
  2. This explanation has been added to the end of "3.3.2. Potential Applications as Thermal-Induced Microgrippers".
  3. The proposed grippers are designed with multiple joints. Since the joints can be triggered by temperature one-by-one, the overall grabbing performance can be controlled. The maximum force is obtained as approximately 0.6 N.
  4. & 5. We thank the reviewer for the suggested reference that has been reviewed and cited in the revised manuscript. The suggested discussion on the computational model has been added to "3.1. Results for the Chiral Tubes".

Reviewer 2 Report

The work is well organized and it offers an interesting point of view on the realization of micro grippers exploiting thermal mechanical metamaterials. The work is a numerical work and I believe that it has to be highlighted just in the begin of the abstract, honestly I thought that this work was both numerical and experimental. 

I think that it could be useful in the introduction to discusse other methods to obtain grippers, for example using elastomeric polymer that response to an external light as reported here: 

1 Advanced Materials 29 (42), 1704047

2 Light: Science & Applications 4 (4), e282-e282

3 Advanced Optical Materials 6 (14), 1800207

This way could be useful in medical/surgical equipments where for example you can attach the micro-grippers on a endoscope tip.

Fix in the main text some issues about the style, there are some word that are smaller than the other ones. 

Figure 8 is not discussed in the text, please introduce a discussion about it and it meaning. 

Author Response

Comment 1

The work is well organized and it offers an interesting point of view on the realization of micro grippers exploiting thermal mechanical metamaterials. The work is a numerical work and I believe that it has to be highlighted just in the begin of the abstract, honestly I thought that this work was both numerical and experimental.

Response

We agree with the reviewer and have modify the Abstract and Introduction to avoid the possible confusion in the revised manuscript. The changes are highlighted in yellow.  

Comment 2

I think that it could be useful in the introduction to discusse other methods to obtain grippers, for example using elastomeric polymer that response to an external light as reported here:

1 Advanced Materials 29 (42), 1704047

2 Light: Science & Applications 4 (4), e282-e282

3 Advanced Optical Materials 6 (14), 1800207

This way could be useful in medical/surgical equipments where for example you can attach the micro-grippers on a endoscope tip.

Response

We thank the reviewer for the suggested references that have been reviewed and cited in the revised manuscript. The discussion has particularly been added to "2.1.2. Origami Microgrippers". The changes are highlighted in yellow.

Comment 3

Fix in the main text some issues about the style, there are some word that are smaller than the other ones.

Response

We have corrected the format issues in the revised manuscript.

Comment 4

Figure 8 is not discussed in the text, please introduce a discussion about it and it meaning.

Response:

Fig. 8 has been described and discussed in the revised manuscript. The changes are highlighted in yellow.

Reviewer 3 Report

The article is well prepared and of scientific interest. I think it can be published.

Author Response

Comment 1

The article is well prepared and of scientific interest. I think it can be published.

Response

We thank the reviewer for the recommendation.

Reviewer 4 Report

In the manuscript entitled "Self-Triggered Thermomechanical Metamaterials with Asymmetric Structures for Programmable Response under Thermal Excitations", the autors focus on two prospective thermomechanical microstructures: the chirally assembled microtubes and thermally-actuated microgrippers (a.k.a. origami microgrippers).

Unfortunately, the authors have confined most of their attention to the microgrippers without having a closer look at the microtubes. The authors do not demonstrate graphically how the chiral tubes behave upon heating/cooling. Overall, the function of chiral tubes as thermally-responsive metamaterials has not been characterized. The role of asymmetry mentioned in the title has not been described at all.

There are virtually no references to works where all the theory presented in Section 2.1 has been derived from. Quality assurance/quality control of the simulations have not been addressed, either.

There are some questions and comments regarding specific details (yet, not much important in view of the overall technical deficiencies).

92-95: Why do the authors use width and height of the hexagonal units when they are equivalent for regular hexagons? How can an angle (θ) denote centroids of the hexagons?

239-240: What is meant by "environmental temperature that reflects human health" so that "the deformation can be used to diagnose human health"?

Furthermore, I have encountered some problems with the language usage, and I suggest that the manuscript should be revised in terms of the correct English.

Please avoid the tautology like "thermally-responsive response".

"Creases width" should be replaced with "crease width".

128-132: Please revise these two sentences for consistence.

164-165: The sentence is unfinished.

Overall, although there are a lot of beautiful demonstrative illustrations and explanatory tables in the manuscript, I do not feel it merits publication in its present form due to the above- mentioned reasons.

Author Response

Comment 1

Unfortunately, the authors have confined most of their attention to the microgrippers without having a closer look at the microtubes. The authors do not demonstrate graphically how the chiral tubes behave upon heating/cooling. Overall, the function of chiral tubes as thermally-responsive metamaterials has not been characterized.

Response

The suggested comments on the chiral tubes have been added to “3.1 Results for the Chiral Tubes” in the revised manuscript. The changes are highlighted in yellow.

Comment 2

The role of asymmetry mentioned in the title has not been described at all.

Response

The discussion on the role of asymmetry has been added in "2.1.1. Chiral Tubes" and "2.1.2. Origami Microgrippers". The changes are highlighted in yellow.

Comment 3

There are virtually no references to works where all the theory presented in Section 2.1 has been derived from. Quality assurance/quality control of the simulations have not been addressed, either.

Response

The suggested references have been added to the theoretical models in Section 2.1 in the revised manuscript. The numerical models in this study are developed using Abaqus. Discussion on the details and validation of the FE models are provided in Section 2.2 in the revised manuscript. The changes are highlighted in yellow.

Comment 4

There are some questions and comments regarding specific details (yet, not much important in view of the overall technical deficiencies).

92-95: Why do the authors use width and height of the hexagonal units when they are equivalent for regular hexagons? How can an angle (θ) denote centroids of the hexagons?

239-240: What is meant by "environmental temperature that reflects human health" so that "the deformation can be used to diagnose human health"?

Response

In response to the concerns of the reviewer:

  • 92-95 in the initial manuscript and xxx-xxx in the revised manuscript: The theoretical model in “2.1.1. Chiral Tubes” have been modified and expanded, adding detailed derivation while explaining the used parameters.
  • 239-240 in the initial manuscript and xxx-xxx in the revised manuscript: We have changed and re-explained the reported chiral stents in "3.3.1. Potential Applications as Chiral Stents". The changes are highlighted in yellow.

Comment 3

Furthermore, I have encountered some problems with the language usage, and I suggest that the manuscript should be revised in terms of the correct English.

  • Please avoid the tautology like "thermally-responsive response".
  • "Creases width" should be replaced with "crease width".
  • 128-132: Please revise these two sentences for consistence.
  • 164-165: The sentence is unfinished.

Response

  • We have simplified the description to “thermal response” in the revised manuscript.
  • "Creases width" has been changed to "crease width" in the revised manuscript.
  • 128-132 in the initial manuscript and xxx-xxx in the revised manuscript: This are relations between the folding angle, crease width, and crease thickness. In particular, the folding angle is proportional to the crease width and inversely proportional to crease thickness.
  • 164-165 in the initial manuscript and xxx-xxx in the revised manuscript: This sentence has been deleted in the revised manuscript.

The changes are highlighted in yellow in the revised manuscript.

Comment 4

Overall, although there are a lot of beautiful demonstrative illustrations and explanatory tables in the manuscript, I do not feel it merits publication in its present form due to the above- mentioned reasons.

Response

We thank the reviewer for the helpful comments and have revised the manuscript accordingly.

Reviewer 5 Report

The authors demonstrate the relevance of the asymmetric self-triggered thermomechanical metamaterials (ST-TM) for the programmable mechanical response subjected to the external thermal field. Numerical models have been investigated to analyze the negative Poisson’s ratio for the chiral tubes and the temperature induced grabbing response for the origami microgrippers. I found the results of the manuscript very interesting. The manuscript is well written and clearly organized. I recommend the publication of the work after addressing the following comments:

1- The color bars in Fig. 6d and 7 are missing.

2- More details related to the numerical simulations performed by Abaqus must be provided so that the results become reproducible.

3- The authors have neglected losses in their simulations. What would happen if they included losses by for instance considering a complex-valued intensity for the governing materials?

4- The bibliography of the manuscript does not look complete to me. It would be great if the authors included some recent review papers about the recent advances in the area of phononic metamaterials such as Reviews in Physics 4 (2019): 100031, Nature Reviews Materials 1.3 (2016): 1-13.

Author Response

Comment 1

The color bars in Fig. 6d and 7 are missing.

Response

The missing information on the color bars have been added in Figs. 6(d) and 7 in the revised manuscript. The changes are highlighted in yellow.

Comment 2

More details related to the numerical simulations performed by Abaqus must be provided so that the results become reproducible.

Response

The suggested details on the numerical simulations have been added in Section 2.2 in the revised manuscript. The changes are highlighted in yellow.

Comment 3

The authors have neglected losses in their simulations. What would happen if they included losses by for instance considering a complex-valued intensity for the governing materials?

Response

We agree with the reviewer that it is helpful to consider losses in the numerical modelling. A discussion on the losses caused by complex-valued intensity has been added to the Conclusions in the revised manuscript. The changes are highlighted in yellow.

Comment 4

The bibliography of the manuscript does not look complete to me. It would be great if the authors included some recent review papers about the recent advances in the area of phononic metamaterials such as

  • Reviews in Physics 4 (2019): 100031,
  • Nature Reviews Materials 1.3 (2016): 1-13.

Response

We thank the reviewer for the helpful references and have reviewed them in the Introduction. The changes are highlighted in yellow.

Round 2

Reviewer 1 Report

Dear authors,

thank you for your rebuttal. The study is significantly improved and only minors need to be resolved.

Please clarify that all parts of figure 8 are your own, otherwise provide a reference.

Reference [43] contains errors in author names, please correct them.

Author Response

Comment 1

Please clarify that all parts of figure 8 are your own, otherwise provide a reference.

Response

We thank the reviewer for the helpful comments. All the materials in Figure 8 are drawn by us.

Comment 2

Reference [43] contains errors in author names, please correct them.

Response

We thank the reviewer for the suggestion. The changes are highlighted in yellow.

Reviewer 4 Report

In the revision, the authors have essentially reworked the manuscript, and in my opinion, this revision has done a lot of good for it. The theory for simulation of the chiral tubes and intelligible results for these structures have been added. The authors have answered my questions raised in my previous review, and I am satisfied with the replies.

However, the manuscript still contains a number of defects. First and formost, it requires language polishing to ensure the correct syntax and proper choice of words, and I recommend its copyediting by an English-speaking expert. For example, the language flaws are noticed in the following lines:

31("are remained"),

60-61 ("antitumor" without object),

71-72, 342, 364 ("outlook" is awkward as a verb),

89 ("typical")

93 ("chiral elements hexagonal cells are rotate"),

189-190 ("that each one"),

191 ("as"),

229, 233 ("that"),

271, 273, 279 ("creases"),

298 ("resulted"),

313 ("perform"),

327 ("obstacle"),

72, 296, 310, 330, 331, 337, 341 ("thermal-induced"),

334 ("under"),

362 ("under temperature"),

364 ("week"),

366 ("debut").

There are a misplaced line break (25) and text in different font sizes (37).

In lines 76-77 and 81-82 the same idea is repeated twice.

In 165, "editable" does not seem to  be an approptriate word.

195: What is meant by "Penultimate “finger”"?

290: "the width from 4, 6, 8, 10, 12, 14 to 16, and" - Which units are used?

291:  What is "the folding number"?

322-326: The sentence is lexically incorrect and difficult to read. 

348:  "the single-chip microcomputer is also widely used" - Something is wrong in the structure of this statement.

362: "the material method" - This is not clear and should be reworded.

"Thermal expansion materials" should be replaced with a more felicitous collocation.

Throughout the manuscript, the authors mention only the axial expansion of the chiral tubes upon heating. However, in Fig. 4, I see that the tubes expand both axially and radially when heated, which fact has not been properly emphasized, graphically presented, and discussed.

Furthermore, in Section 3.3.1, the authors propose chiral tubes as indicators of human temperature embedded in human body and coupled with piezoelectric or triboelectric materials to convert the thermal deformation into electric signal. This possible application does not seem realistic to me for two reasons. First, it is difficult to find or prepare an immunoneutral metamaterial. Second, transducer materials embedded together with chiral tubes into human body require some wiring that will need to pass through human tissues from the inside out, which is neither feasible nor safe.

Thus, in my opinion, the manuscript is not fully ready for publication, and I recommend its revision.

Author Response

Comment 1

However, the manuscript still contains a number of defects. First and formost, it requires language polishing to ensure the correct syntax and proper choice of words, and I recommend its copyediting by an English-speaking expert. For example, the language flaws are noticed in the following lines:

  • 31("are remained"),
  • 60-61 ("antitumor" without object),
  • 71-72, 342, 364 ("outlook" is awkward as a verb),
  • 89 ("typical")
  • 93 ("chiral elements hexagonal cells are rotate"),
  • 189-190 ("that each one"),
  • 191 ("as"),
  • 229, 233 ("that"),
  • 271, 273, 279 ("creases"),
  • 298 ("resulted"),
  • 313 ("perform"),
  • 327 ("obstacle"),
  • 72, 296, 310, 330, 331, 337, 341 ("thermal-induced"),
  • 334 ("under"),
  • 362 ("under temperature"),
  • 364 ("week"),
  • 366 ("debut").
  • There are a misplaced line break (25) and text in different font sizes (37).
  • In lines 76-77 and 81-82 the same idea is repeated twice.
  • In 165, "editable" does not seem to be an approptriate word.
  • 195: What is meant by "Penultimate “finger”"?
  • 290: "the width from 4, 6, 8, 10, 12, 14 to 16, and" - Which units are used?
  • 291:  What is "the folding number"?
  • 322-326: The sentence is lexically incorrect and difficult to read. 
  • 348:  "the single-chip microcomputer is also widely used" - Something is wrong in the structure of this statement.
  • 362: "the material method" - This is not clear and should be reworded.
  • "Thermal expansion materials" should be replaced with a more felicitous collocation.

Response

We thank the reviewer for the helpful comments and have revised the manuscript accordingly. The changes are highlighted in yellow. In response to the concerns of the reviewer:

195 in the initial manuscript and 193 in the revised manuscript: We have corrected the "Penultimate “finger”" to “last joint”.

291 in the initial manuscript and 289 in the revised manuscript: “the folding number” described here means that the number of joints which have folded.

Comment 2

Throughout the manuscript, the authors mention only the axial expansion of the chiral tubes upon heating. However, in Fig. 4, I see that the tubes expand both axially and radially when heated, which fact has not been properly emphasized, graphically presented, and discussed.

Response

We thank the reviewer for the suggestions. We agree that the chiral tubes do experience thermal expansion in the axial and radical directions. However, this paper focuses on the rotation angle of the hexagonal cells and the axial stretching of the entire tubes. As a consequence, the radical expansion has not been particularly included and discussed.

Comment 3

Furthermore, in Section 3.3.1, the authors propose chiral tubes as indicators of human temperature embedded in human body and coupled with piezoelectric or triboelectric materials to convert the thermal deformation into electric signal. This possible application does not seem realistic to me for two reasons. First, it is difficult to find or prepare an immunoneutral metamaterial. Second, transducer materials embedded together with chiral tubes into human body require some wiring that will need to pass through human tissues from the inside out, which is neither feasible nor safe.

Response

We thank the reviewer for proposing the question. For the first question, we can take the approach of wrapping chiral stents in the immunoneutral metamaterial. For the second question, we can use ultrasonic transducers to convert electrical signals into ultrasound, which is a way of achieving wireless. In addition, the application we proposed are potential and provides ideas for future related research.

Reviewer 5 Report

I recommend the publication of the work at this stage.

Author Response

We thank the reviewer for the recommendation.